# ROLL THE DICE: MONTE CARLO DOWNSAMPLING AS A LOW-COST ADVERSARIAL DEFENCE

## ABSTRACT

The well-known vulnerability of Neural Networks to adversarial attacks is concerning, more so with the increasing reliance on them for real-world applications like autonomous driving, medical imaging, and others. Multiple previous works have proposed defense methods against adversarial attacks, including adversarial training, adding random noise to images, frequency pooling, and others. We observe from several such works, that there are two main paradigms for mitigating adversarial attacks. First, effective downsampling leads to learning better feature representations during training, thus improving the performance on attacked and non-attacked samples. However, these methods are expensive. Second, perturbing samples with for example random noise helps in mitigating adversarial attacks as they stymie the flow of gradients to optimize the attacks. However, these methods lower the network's performance on non-attacked samples. Thus, in this work, we combine the best of both strategies to propose a novel Monte-Carlo sampling-based approach for downsampling called Stochastic Downsampling. We combine bi-linear interpolation with Monte Carlo integration for performing downsampling. This helps us mitigate adversarial attacks while preserving the performance of non-attacked samples, thus increasing reliability. Our proposed Stochastic Downsampling operator can easily be integrated into any existing architecture, including adversarially pre-trained networks, with some finetuning. We show the effectiveness of Stochastic Dowsampling over multiple image classification datasets using different network architectures with different training strategies. We provide the code for performing Stochastic Downsampling here: Anonymous GitHub Repository.

## 1 INTRODUCTION

The advent of Machine Learning (ML) methods, specifically in Computer Vision (CV), has fueled their increased application for real-world applications such as Autonomous Driving(Hu et al., 2023), Semantic Segmentation(Ronneberger et al., 2015; Chen et al., 2017), Optical Flow Estimation(Dosovitskiy et al., 2015; Ilg et al., 2017; Teed & Deng, 2020), Panoptic Segmentation(Sirohi et al., 2023; Mohan & Valada, 2021), Image Restoration(Zamir et al., 2022; Chen et al., 2022; Agnihotri et al., 2023a), among others. The reliability of models trained with such methods is of paramount importance for their applications, especially those where human safety is critical. However, prior works(Goodfellow et al., 2015; Kurakin et al., 2017; Gu et al., 2022; Schrodi et al., 2022; Agnihotri et al., 2023c; Grabinski et al., 2022b; 2023; Croce et al., 2021; Hendrycks & Dietterich, 2019; Wong et al., 2020) have shown that ML methods are susceptible to adversarial attacks and distribution shifts making them non-robust. These vulnerabilities of a non-robust ML model can be exploited by an attacker to fool the model, or by natural weather conditions, to fail the model on the target task. This adversely affects their reliability for any safety critical real-world task.

Prior works have proposed methods to alleviate this vulnerability by either encouraging the ML model to learn more stable feature representations or by obstructing the optimization process of the attacks, such that the attack effectiveness is reduced. The former can be achieved by either using learning strategies, such as adversarial training (Salman et al., 2020; Liu et al., 2023; Singh et al., 2024) or by architectural design choices (Grabinski et al., 2022a; 2023; Agnihotri et al., 2023b) that lead to the ML model learning better representation thus increasing their endurance of attacks and distribution shifts. The latter can be achieved by adding blurring (Zhang, 2019) or noise (Zhang,

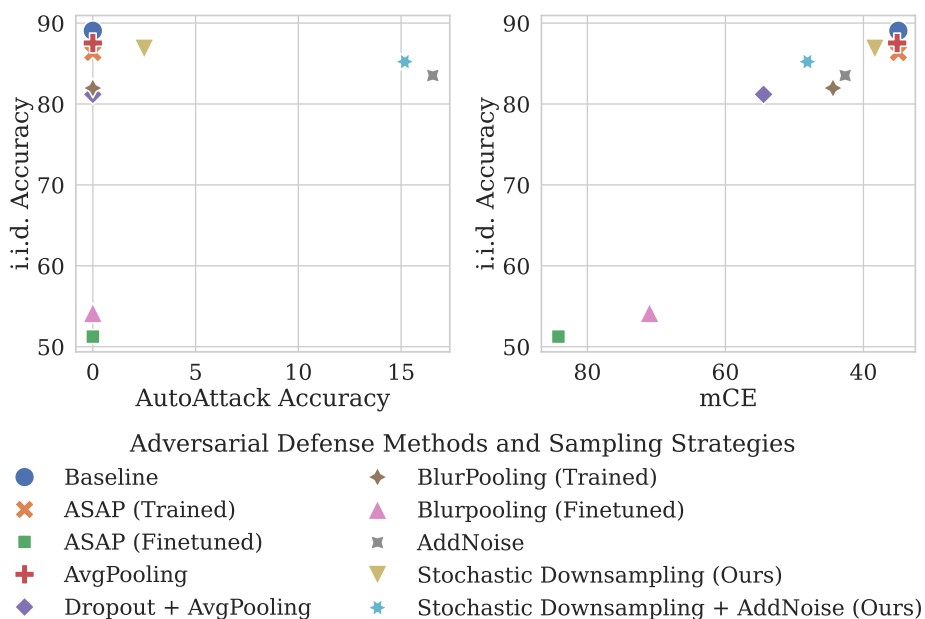

Figure 1: Comparing the performance of various adversarial attack defense methods and downsampling approaches intended to improve robustness. We observe that our proposed Stochastic Downsampling offers the best trade-off between i.i.d. accuracy, reliability, and generalization.

2019; Rony et al., 2019) to the feature maps. However, on the one hand, the proposed learning strategies require training the ML models (such that the epochs of training the robust ML model are equal to the epochs of training required by the non-robust model), often from scratch which requires significant computation, especially increasing the time complexity. These methods do not consider the information already learned by the non-robust model. Training the ML models with adversarial attacks (Kurakin et al., 2017; Wong et al., 2020) has the added complexity of performing the attacks during training which can be very expensive (Agnihotri et al., 2023c). On the other hand, strategies like adding noise to disturb the gradient flow also corrupt the image and therefore lead to a loss in performance on independent and identically distributed (i.i.d.) samples (not adversarially attacked) of images and also lead to a loss in generalization, for example, to changes in distribution due to weather conditions or digital corruptions. Additionally, these methods also fail to protect the model against attacks that do not require the passing of gradients through the model for optimizing the attack, i.e. black-box adversarial attacks like Squares attack(Andriushchenko et al., 2020) in AutoAttack(Croce & Hein, 2020c).

To address this issue, we propose **Stochastic Downsampling (SD)**, an adversarial defense method that helps the model be robust against adversarial attacks and common corruptions(Hendrycks & Dietterich, 2019) while preserving the performance of the model on clean samples. Unlike existing gradient obfuscation defenses, it provides robustness against zero-order (black-box) adversarial attacks due to its inherent stochasticity. At the same time, the sampling in the proposed operation is purely done within the variance of the existing data, allowing it to be used within pre-trained models with only little adaptation and to perform at very low cost in terms of clean accuracy.

Specifically, Stochastic Downsampling changes the downsampling operation in a ML model, replacing it with a Monte-Carlo Integration[1], followed by bilinear interpolation. For Convolutional Neural Networks (CNNs), this is achieved by changing the stride of the strided convolution operations used for downsampling to one and appending the new Stochastic Downsampling layer to it for downsampling feature maps. Architectures like ViT (Touvron et al., 2021; Tan & Le, 2021; Radosavovic et al., 2020) do not have a downsampling step and thus might require a different approach. Our proposed Stochastic Downsampling has no additional learnable parameters and thus does not require learning. However, the other model weights might require some finetuning to adapt

---

[1]please refer to (Caflisch, 1998) for more details on Monte Carlo Integration

to the new downsampling method and thus are trained with a low learning rate for very few epochs (specifically just 5 epochs) providing us with a low-cost solution. The Monte-Carlo Integration provides the stochasticity required to disorient the attacks, while the finetuning helps preserve model performance.

In Fig. 1, we show the gains from Stochastic Downsampling, compared to other approaches, and show that our method provides the best trade-off between i.i.d. performance, reliability (shown by AutoAttack Accuracy), and ability to generalize to image corruptions (show by mean Corruption Error i.e. mCE). We describe the method in detail in Section 3.

The following are the most important contributions of our work:

- We propose a novel downsampling operation Stochastic Downsampling that provides defense against adversarial attacks without any additional learnable parameters.
- Our method preserves most of the i.i.d. performance of the model while helping improve reliability under adversarial attacks.
- Stochastic Downsampling can be included in the model architecture with elementary and straightforward modifications.
- We provide an in-depth analysis of our proposed method in comparison to other methods and show that Stochastic Downsampling offers the best possible trade-off.

## 2 RELATED WORK

Following, we discuss prior works done toward defense from adversarial methods

**Gradient Obfuscation** All white-box adversarial attacks attempt to optimize the attack noise by back-propagating the loss gradients to the input image. However, if this flow of gradients were to be disturbed, it would interfere with the optimization ability of the attack and thus such methods would be hacks that work at adversarial defense. This is known as "Obfuscated Gradients" as shown by Athalye et al. (2018). They categorized obfuscation of gradients into three types, *Shattered Gradients*, where the gradients are incorrect or non-existent, *Exploding & Vanishing Gradients*, and *Stochastic Gradients*, which causes incorrect estimation of the gradients. Our proposed Stochastic Downsampling might appear similar to the Stochastic Gradient type of Obfuscated Gradients method, however, as we sample multiple points from the valid data space (i.e. within the variance of correct sampling), we simply change the direction of the gradients within their correct range rather than making them incorrect. Thus, Stochastic Downsampling is more than just a gradient obfuscation method but is an efficient sampling method with stochasticity.

Byun et al. (2022); Nguyen et al. (2023); Li et al. (2019) proposed adding noise at various stages of the model. For our comparative analysis, we take inspiration from them and include "AddNoise", as a method for comparison. Here, in each forward pass on the model, we add noise to feature maps, after downsampling them. The noise itself can be sampled from a Gaussian or a uniform distribution and has the same spatial resolution as the feature maps to which it is added.

While moderately effective in disturbing gradient flow and thus weakening the adversarial attacks, these methods lead to a significant drop in clean performance. This is explained by Zhang et al. (2019) and Tsipras et al. (2019), which show there exists a trade-off between robustness and clean performance of a model. However, we demonstrate that Stochastic Downsampling can achieve a significantly better trade-off than some simple hacks, helping the model extract meaningful representations during downsampling.

**Adversarial Training** Adversarial training is one of the most promising methods to enhance the model's robustness, especially in the presence of adversarial attacks (Goodfellow et al., 2015; Kurakin et al., 2017; Moosavi-Dezfooli et al., 2016; Carlini & Wagner, 2017; Rony et al., 2019) but also to enhance general model robustness (Croce et al., 2020). During adversarial training, the model is confronted with adversarial samples by adding an additional loss term (Liu et al., 2023; Kurakin et al., 2017), showing augmented inputs (Geirhos et al., 2018) or adding additional external or generated inputs. One widely used additional data source is using *ddpm* (Gowal et al., 2021; Rade & Moosavi-Dezfooli, 2021; Rebuffi et al., 2021) dataset which is generated by Ho et al. (2020) and

includes one million additional samples for CIFAR-10. For the evaluation and collection of robust models RobustBench (Croce et al., 2020) provides a compressive overview of recent adversarially trained models and their performance on AutoAttack (Croce & Hein, 2020a) and Common Corruptions (Hendrycks & Dietterich, 2019; Kar et al., 2022).

However, most of these methods to enhance the model's robustness rely heavily on additional data or data augmentation which takes much longer to train. In the case of traditional adversarial training, one needs even several forward and backward passes to calculate the adversarial noise depending on the adversarial attack used to generate the perturbations. Perturbations generated using several iterations (Kurakin et al., 2017) provide stronger robustness than perturbations generated with a single iteration (Goodfellow et al., 2015). Summarizing, adversarial training mostly comes at an increased amount in computations needed due to more samples and a harder learning problem this can increase the training time by a factor between seven and fifteen (Kurakin et al., 2017; Wang et al., 2020; Wu et al., 2020; Zhang et al., 2019; Grabinski et al., 2022a)

**Anti-Aliasing Sampling for increased Robustness**   Prior works on inherently improving robustness via Anti-Aliasing Sampling include aliasing-free downsampling like Frequency Low Cut (FLC) Pooling (Grabinski et al., 2022a), aliasing- and sinc-artifact-free pooling (ASAP) (Grabinski et al., 2023), BlurPooling (Zhang, 2019) or adaptive BlurPooling (Zou et al., 2020). While BlurPooling and adaptive BlurPooling use blurring before downsampling to reduce aliasing and ensure greater shift invariance, FLC Pooling and ASAP ensure aliasing-free downsampling, leading to higher native robustness and a reduced risk of catastrophic overfitting in FGSM adversarial training. In Grabinski et al. (2022b), the authors show a strong negative correlation between aliasing after downsampling and the robustness of a network. Thus, ensuring aliasing-free downsampling increases a network's robustness.

# 3 METHOD

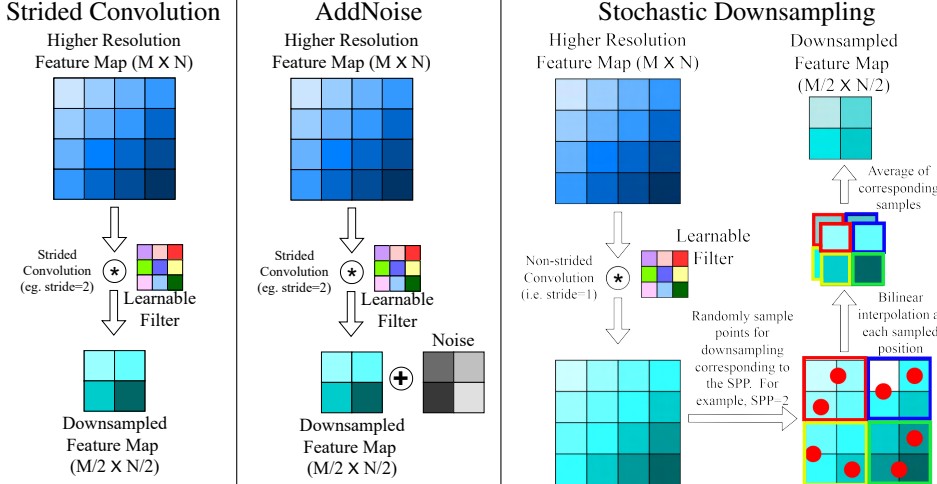

Figure 2: An Abstract representation of downsampling operations performed by strided convolution, AddNoise, and Stochastic Downsampling.

The rise of Deep Learning in computer vision is undoubtedly an impressive achievement. There are several modifications and adjustments that keep developing the field in the domain such as object detection, segmentation, and so on. One such development is the modification of the Pixel lattice structure in the sensor by Sommerhoff et al. (2023). The proposed idea of a differential sensor simulation framework modified the pixel lattice using rectilinear and curvilinear deformation. This helped the model to capture better feature representation when the image is downsampled using the deformed sensor layout.

The practical implementation of Sommerhoff et al. (2023) aims to efficiently compute the accumulated incoming radiance $L_i$ captured by a (non-uniform) sensor pixel as

$$I_k = \frac{1}{\text{area}(A_k)} \int_{A_k} W(x) L_i(x) dx, \tag{1}$$

where $A_k$ is the set containing every point of the $k$-th pixel, $W$ is a weighting function that can model spatially varying pixel response and $I_k$ is the final pixel color.

The analytic integral in Eq. 1 can be approximated by Monte Carlo integration:

$$I_k \approx \frac{1}{n} \sum_i^n W(x_i) L_i(x_i), \tag{2}$$

where $x_i$ are uniform random samples inside the pixel. Computing this integral with Monte Carlo sampling is similar to stochastic multisampling as a spatial anti-aliasing technique, which is commonly utilized in computer graphics, especially for photorealistic path-tracing (Ernst et al., 2006; Glassner, 2014; Pharr et al., 2024).

The quality of this approach scales with the number of random samples per pixel and the expected value is the true integral. On the contrary a lower number of samples results in higher variance and thus more noise. We ablate over the choice of *Samples Per-Pixel (SPP)*, to find an ideal number.

Since a closed form expression for the incoming radiance $L_i$ is generally not available and simulation, e.g. by raytracing, is computationally expensive, Sommerhoff et al. (2023) propose to approximate $L_i$ by existing high-resolution images. These images can be sampled at arbitrary positions by bilinear interpolation. Together with Monte Carlo integration, this effectively results in a *Stochastic Downsampling operation*, if the resolution of the target sensor is lower than the resolution of the high-resolution input image.

We make the observation that this downsampling scheme can be naturally extended from images to more general feature maps $F \in \mathbb{R}^{W \times H \times C}$. For this, we make the simplifying assumptions of uniform pixels and constant $W(x) = 1$. In the following, square brackets denote querying a feature map at integer locations, i.e. $F[i,j] \in \mathbb{R}^C$, whereas parenthesis denotes bilinear interpolation of the for nearest neighbors at not necessarily integer coordinates, e.g. $F(x,y) \in \mathbb{R}^C$. Using this notation, our stochastic downsampling operation in total can thus be expressed as shown in Eq. (3),

$$F \downarrow_\alpha [u,v] = \frac{1}{n} \sum_i^n F(\alpha x_i, \alpha y_i) \tag{3}$$

where $x_i \sim \mathcal{U}_{[u,u+1]}$ and $y_i \sim \mathcal{U}_{[v,v+1]}$ follow uniform distributions inside the current pixel.

The **Stochastic Downsampling** operation can replace other pooling operations like average pooling, or downsampling operations like strided convolution, commonly used in many neural network architectures. We implement it in PyTorch using the grid_sample function, which is differentiable with respect to the input feature map.

Most architectures use a strided convolution for downsampling, as shown by Eq. (4) for a downsampling factor of two.

$$\sum_{2m}^M \sum_{2n}^N F_{M \times N}(x_m, y_n) K_{i \times j}(i-m, j-n) = F_{\frac{M}{2} \times \frac{N}{2}} \tag{4}$$

where $K$ is the learned convolution kernel. In our modification, instead of taking a step of size two in Eq. (4), we modify the step size to one and then use Eq. (3) to perform the downsampling operation.

To perform ablation studies, we experiment with a few other methods, we describe them here:

**AvgPool:** We change the stride in Eq. (4) to one, and use Average Pooling with a $2 \times 2$ kernel for downsampling.

**DropOut + AvgPool:** To ablate over the repercussions of looking at merely 2 pixels (at random), when downsampling with a $2 \times 2$ sized kernel in Average Pooling, we use DropOut(Srivastava et al., 2014) with a dropping probability of 50% after convolution with a stride of one, and before the AvgPool operation.

**AddNoise:**     To ablate if our gains are merely due to adding noise to feature maps, or truly due to the Stochastic Downsampling operation itself, we perform downsampling as shown by Eq. (4), but add noise to the feature maps, after the downsampling. The noise can be sampled from a uniform distribution or a Gaussian distribution.

We provide an abstract overview of the prominent downsampling methods in Fig. 2.

## 4 EXPERIMENTS

Following, we report the implementation details of the experiments performed and discuss the observations made on the results.

### 4.1 EXPERIMENTAL SETUP

Here we provide an overview of the implementation details, we provide additional implementation details in Appendix A.

**Adversarial Attacks.** We use PGD(Kurakin et al., 2017), APGD(Wong et al., 2020) and AutoAttack(Croce & Hein, 2020a), with $\ell_\infty$-norm bounded $\epsilon = \frac{4}{255}$ and $\alpha$=0.01 for all our experiments. PGD and APGD are white-box attacks, meaning they require an undisturbed flow of gradients for optimizing their attack. However, AutoAttack, as proposed comprises APGD-CE (non-targeted APGD attack with Cross Entropy loss), APGD-T (targeted APGD attack), FAB(Croce & Hein, 2020b) and Square(Andriushchenko et al., 2020) Attacks, from these, Square Attack is a black-box attack that does not require a flow of gradients through the ML model. Additionally, since Stochastic Downsampling is essentially a gradient obfuscation method, we also test against Square Attack (Andriushchenko et al., 2020) alone, as it is a black-box attack and does not use the gradient information of a model to optimize the attack.

**Metrics.** For independent and identically distributed (i.i.d.) (non-attacked and non-perturbed) samples, we report the i.i.d. Accuracy (i.i.d. Acc). For evaluations against adversarial attacks, we report the accuracy after the respective attack. For samples from the 2D Common Corruptions variant of the respective datasets, we report the mean Corruption Error (mCE), this is the mean error by the method on all the corrupted samples. All numbers are reported in percentages.

A high i.i.d. accuracy indicates good performance, while a high accuracy against adversarial attacks indicates more reliability and a lower mCE value indicates more generalization ability.

### 4.2 RESULTS

Following we report the experimental results, comparing our proposed Stochastic Downsampling (SD), with other known methods which can be used for defense against adversarial attacks.     In

Table 1: Here we report the performance of various defense methods against adversarial attacks and common corruptions using ConvNeXt-tiny and the **ImageNet100 dataset**. We perform these experiments over three different seeds and report the mean and standard deviation (std) as 'mean±std'. [†] denotes longer training until convergence of training loss. All other methods are finetuned for merely five epochs.

| Defense Method | i.i.d. Acc. (%)↑ | PGD Acc. (%)↑ | AutoAttack Acc. (%)↑ | Square Attack Acc. (%)↑ | mCE (%)↓ |
|---|---|---|---|---|---|
| Baseline[†] | 89.05 ± 0.1 | 0.51 ± 0.06 | 0 ± 0 | 36.26 ± 0.54 | 34.94 ± 0.31 |
| ASAP[†] | 86.36 ± 0.21 | 1.91 ± 0.2 | 0 ± 0 | 13.59 ± 0.65 | 34.94 ± 0.53 |
| ASAP | 51.25 ± 1.12 | 0.01 ± 0.01 | 0 ± 0 | 5.93 ± 0.21 | 84.2 ± 0.89 |
| AvgPool | 87.54 ± 0.16 | 1.49 ± 0.24 | 0 ± 0 | 24.07 ± 0.47 | 35.11 ± 0.49 |
| DropOut + AvgPool | 81.19 ± 0.71 | 0.42 ± 0.04 | 0 ± 0 | 12.66 ± 0.61 | 54.47 ± 1.69 |
| Blurpooling[†] | 81.97 ± 0.13 | 0.57 ± 0.25 | 0 ± 0 | 12.53 ± 0.86 | 44.41 ± 0.52 |
| Blurpooling | 54.17 ± 3.3 | 0.23 ± 0.03 | 0.01 ± 0.01 | 3.79 ± 0.42 | 70.99 ± 2.54 |
| AddNoise (uniform) | 87.25 ± 0.15 | 33.49 ± 0.81 | 1.78 ± 0.11 | 85.33 ± 0.19 | 37.98 ± 0.11 |
| AddNoise (std=0.75) | 83.53 ± 0.05 | 40.55 ± 0.55 | 16.4 ± 0.18 | 79.78 ± 0.49 | 42.96 ± 0.29 |
| SD (Ours) | 86.83 ± 0.16 | 40.81 ± 0.64 | 2.5 ± 0.1 | 83.08 ± 0.07 | 38.34 ± 0.6 |
| SD + AddNoise(std=0.15) (Ours) | 85.23 ± 0.24 | 48.12 ± 0.14 | 15.18 ± 0.1 | 81.3 ± 0.16 | 39.24 ± 0.25 |

Tab. 1, we observe that methods such as BlurPooling and ASAP require long training and do not perform well when simply fintuned at a low budget. The recently proposed ASAP significantly

outperforms BlurPooling in all aspects, i.e. i.i.d. accuracy, OOD robustness and adversarial robustness. Whereas, certain variants of AddNoise outperform ASAP w.r.t. adversarial robustness, while ASAP still outperforms Addnoise variants in i.i.d. performance and OOD robustness. Please note, here AddNoise variants were only finetuned for 5 epochs, whereas ASAP requires full training. AddNoise and ASAP provide a trade-off, here we trade i.i.d. performance and generalization ability with reliability under adversarial attacks. However, this is not ideal, we require our models to have good i.i.d. performance, generalization ability, and reliability. To this effect, Stochastic Downsampling comes in handy. As shown in Tab. 1, Stochastic Downsampling provides the best possible trade-off, for an insignificant drop in i.i.d. accuracy, and generalization ability, it provides significant gains in reliability under adversarial attacks. In case of scenarios where adversarial robustness is more important, Stochastic Downsampling can also be coupled with AddNoise to trade-off some i.i.d. accuracy and OOD robustness for more adversarial robustness.

Stochastic Downsampling might be considered similar to a gradient obfuscation method (Athalye et al., 2018). Thus, to ascertain that it is *not providing a false sense of security*, we additionally perform Square Attack, a black-box adversarial attack that does not require gradient information of the model to optimize the attack. We observe that the performance of the model with Stochastic Downsampling is almost unaffected under Square attack. This shows that Stochastic Downsampling is not providing a false sense of security.

## 5 ANALYSIS AND ABLATION STUDIES

Following we provide analysis and ablation studies to demonstrate that despite being similar to a gradient obfuscation method, Stochastic Downsampling does not provide a false sense of security. Additionally, we demonstrate the versatility and ease of use of Stochastic Downsampling.

### 5.1 EXTENDING TO OTHER MODELS AND DATASETS

The gains obtained using Stochastic Downsampling are not limited to the ConvNeXt-tiny model and ImageNet100 dataset but extend to other models and larger datasets as well. To demonstrate this we extend the experiments to ConvNeXt-Small, ConvNeXt-Base, ResNet18, ResNet50, and ResNet101 on the ImageNet-1k dataset. These models were pretrained on the ImageNet-1k dataset, and then

Table 2: Here we report the performance of various model finetuning strategies against adversarial attacks for the **ImageNet-1k dataset**. All methods except 'Baseline' are finetuned for 5 epochs.

| Model | Method | i.i.d. Accuracy (%)↑ | PGD Acc (%)↑ | AutoAttack Acc. (%)↑ |
|---|---|---|---|---|
| ConvNeXt-T | Baseline | 82.06 | 1.08 | 0.00 |
| | SD + AddNoise | 77.55 | 29.05 | 6.00 |
| | SD | 79.21 | 20.25 | 0.94 |
| ConvNeXt-S | Baseline | 83.15 | 3.7 | 0.00 |
| | SD + AddNoise | 79.23 | 32.34 | 4.86 |
| | SD | 80.45 | 23.59 | 0.56 |
| ConvNeXt-B | Baseline | 83.83 | 5.12 | 0.00 |
| | SD + AddNoise | 80.01 | 29.98 | 3.32 |
| | SD | 81.02 | 21.41 | 0.66 |
| ResNet18 | Baseline | 69.76 | 0.29 | 0.00 |
| | SD + AddNoise | 68.08 | 34.41 | 0.14 |
| | SD | 69.24 | 19.39 | 0.80 |
| ResNet50 | Baseline | 76.15 | 1.28 | 0.00 |
| | SD + AddNoise | 73.73 | 53.80 | 0.16 |
| | SD | 75.39 | 34.44 | 1.00 |
| ResNet101 | Baseline | 77.37 | 2.33 | 0.00 |
| | SD + AddNoise | 75.35 | 55.70 | 0.38 |
| | SD | 76.60 | 36.59 | 0.94 |

the strided convolution layers used in them for downsampling were replaced with non-strided Convolution layers (with the same weights as the strided-convolution) and Stochastic Downsampling

layer. The resultant models were then finetuned for 5 epochs. We observe in Tab. 2 that the minimal trade-off between i.i.d. accuracy and adversarial robustness observed in Sec. 4.2 still holds demonstrating the effectiveness of Stochastic Downsampling even with large models on vast datasets with significantly many classes. Please refer to the Tab. 6 for experiments with CIFAR-100.

## 5.2 BETTER UNDERSTANDING THE TRADE-OFF

We observed the performance trade-off by AddNoise, Stochastic Downsampling, and the combination of Stochastic Downsampling and AddNoise in Sec. 4.2. Intruiged by this trade-off we attempt to understand it better and thus perform more detailed evaluations as reported in Tab. 3. Here we

Table 3: Here we report the performance of ConvNeXt-tiny with various defense strategies against AutoAttack for the **ImageNet100 dataset**. The noise for AddNoise is sampled from a normal distribution with mean=0 and different standard deviations (std) denoted below, except AddNoise (Uniform), here the noise is sampled from a uniform distribution. All methods are finetuned for 5 epochs, except those denoted by $^\dagger$, these are finetuned for a longer duration (until training loss converges).

| Method | i.i.d. Accuracy (%)↑ | AutoAttack (%)↑ | | | | mCE (%)↓ |
|---|---|---|---|---|---|---|
| | | APGD-CE | APGD-T | FAB-T | Square | |
| Baseline | 89.00 | 0.00 | 0.00 | 0.00 | 0.00 | 35.00 |
| AddNoise (std=0.05) | 88.82 | 0.38 | 0.14 | 0.06 | 0.02 | 34.842 |
| AddNoise (std=0.10) | 88.48 | 0.56 | 0.26 | 0.06 | 0.06 | 35.381 |
| AddNoise (std=0.15) | 88.12 | 0.92 | 0.36 | 0.24 | 0.12 | 36.217 |
| AddNoise (std=0.30) | 87.18 | 9.02 | 2.94 | 2.76 | 2.72 | 37.879 |
| AddNoise (std=0.50) | 85.68 | 20.22 | 10.66 | 10.28 | 10.26 | 40.218 |
| AddNoise (std=0.75) | 83.52 | 24.82 | 17.02 | 16.74 | 16.54 | 42.65 |
| AddNoise$^\dagger$ (std=0.75) | 86.76 | 1.28 | 0.28 | 0.18 | 0.12 | 37.753 |
| AddNoise (std=0.90) | 82.48 | 26.54 | 18.84 | 18.36 | 18.28 | 43.916 |
| AddNoise (std=1.0) | 81.58 | 26.74 | 19.82 | 19.5 | 19.38 | 44.406 |
| AddNoise (Uniform) | 87.10 | 6.48 | 2.18 | 1.96 | 1.92 | 37.853 |
| SD + AddNoise (std=0.05) | 86.12 | 14.48 | 6.14 | 5.94 | 5.88 | 37.837 |
| SD + AddNoise (std=0.1) | 85.62 | 21.12 | 11.40 | 11.18 | 11.10 | 39.171 |
| SD + AddNoise (std=0.15) | 84.98 | 26.48 | 15.36 | 15.14 | 15.12 | 38.955 |
| SD + AddNoise (std=0.9) | 75.68 | 37.72 | 29.38 | 28.74 | 28.42 | 46.403 |
| SD + AddNoise (std=0.15)$^\dagger$ | 86.76 | 14.82 | 6.00 | 5.80 | 5.74 | 36.941 |
| SD | 86.80 | 7.70 | 3.00 | 2.64 | 2.60 | 37.74 |

vary, the degree of noise added to the feature maps after downsampling. That is for AddNoise when sampling from a Normal distribution, we keep the mean equal to zero and vary the standard deviation from 0.05 to 1.0. Additionally, we do the same when combining Stochastic Downsampling and AddNoise. Then, we arrive at the best possible trade-off, for AddNoise it is with a standard deviation equal to 0.75. For the combination of Stochastic Downsampling and AddNoise, a standard deviation of 0.15 provides a decent trade-off. However, depending on the scenario, one is free to choose their ideal trade-off. As shown in Tab. 3, as the standard deviation increases, the accuracy against adversarial attacks increases, and the i.i.d. accuracy and generalization ability decreases.

## 5.3 TRANSFER ATTACK COMPARISON

Apart from using Square Attacks in Sec. 4.2, to negate the argument of a false sense of security due to gradient obfuscation, we transfer adversarial attacks from the baseline model, which allows the adversarial attack to be optimized without any gradient obfuscation to models that inhibit the free flow of gradients. We report our findings in Tab. 4. Here we observe that the attack is indeed strong against the baseline model of ConvNeXt-tiny. However, when the attack is transferred to models with alleged gradient obfuscation, that is models with AddNoise and Stochastic Downsampling, the strength of the attack is reduced. Moreover, while AddNoise is reducing the attack to a great extent, the model with Stochastic Downsampling is hardly affected by the attack thus reducing its strength even more. This demonstrates that Stochastic Downsampling is not just another gradient obfuscation method but is helping the network defend against adversarial attacks by extracting meaningful representations even under adversarial attacks.

Table 4: Here we report the performance of transfer attacks across different defense strategies with ConvNeXt-tiny using the **ImageNet100 dataset**. Defense methods are evaluated on adversarial samples optimized using the 'Baseline' ConvNeXt-tiny.

| Model | PGD (%)↑ | AutoAttack (%)↑ |
|---|---|---|
| Baseline | 0.54 | 0 |
| AddNoise (std= 0.75) | 45.24 | 45.4 |
| SD | 75.06 | 80.3 |
| SD + AddNoise (std=0.15) | 65.14 | 65.6 |

## 5.4 ABLATING THE EFFECT OF CONTEXT

As discussed in Sec. 3, the Stochastic Downsampling operation samples only two pixels (chosen at random) in a region of the feature maps to downsample to one, i.e. samples per pixel (SPP) is two. However, it would be interesting to ablate this spatial context available to Stochastic Downsampling. Thus, we ablate increasing this spatial context such that the operation samples each downsampled pixel using a varying number of pixels from the higher-resolution feature map. We report our

Table 5: Ablation over different values of samples per pixel (SPP) for Stochastic Downsampling performed using ConvNeXt-tiny and **ImageNet100 dataset**.

| SPP (%)↑ | i.i.d. Accuracy (%)↑ | PGD Accuracy (%)↑ |
|---|---|---|
| 1 | 85.78 | 50.22 |
| 2 | 86.8 | 41.38 |
| 4 | 87.06 | 30.76 |
| 8 | 87.02 | 18.14 |
| 16 | 87.02 | 9.06 |

findings in Tab. 5 and observe that as we increase context the i.i.d. accuracy increases, however, that saturates after four samples per pixel. While the robustness of the model consistently decreases with increasing context. Additionally, we observe that at SPP=2, the i.i.d. accuracy is marginally higher than SPP=1. Thus, we use SPP=2 for our Stochastic Downsampling.

## 6 CONCLUSION

Adversarial attacks pose a threat to Deep Learning based methods. It is of paramount importance that reliable DL-based methods can defend against such threats to a reasonable extent. However, most adversarial defense methods inherently create a challenging trade-off between i.i.d. performance and robustness. Stochastic Downsampling eases this challenge by significantly improving the trade-off by increasing reliability with only a marginal drop in i.i.d. accuracy and generalization ability of DL-based methods. Stochastic Downsampling is easy to incorporate in pre-trained models and requires very limited finetuning to perform at its peak efficiency. The gains from Stochastic Downsampling are consistent across model architectures, and datasets. We show that despite being very similar to a gradient obfuscation based defense method, Stochastic Downsampling does not provide a false sense of security. This work is a step in the direction of sampling-based adversarial defense methods that help extract meaningful representations during downsampling, even under adversarial attacks.

### LIMITATIONS

There is still a marginal drop in clean performance, ideally this should also be avoided. While Stochastic Downsampling mitigates adversarial attacks better than other defense methods, there is still significant room for improvement.

## REPRODUCIBILITY STATEMENT

All experimental results in this work are reproducible. We make the code base available to the reviewers and will make it public upon acceptance. We understand that the evaluations involve stochasticity and thus to demonstrate the effectiveness of our proposed Stochastic Downsampling, we perform multiple experiments over 3 different seeds and report the mean and standard deviation. We observe that the standard deviation is quite low indicating that the improvements due to Stochastic Downsampling cannot be attributed to the stochastic behaviour.

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

# Roll the dice: Monte Carlo Downsampling
# as a low-cost Adversarial Defence

## Paper #1058 Supplementary Material

## A  IMPLEMENTATION DETAILS

Following we provide in-depth implementation details for the experiments.

### A.1  EXPERIMENTAL SETUP

Here we provide an overview of the implementation details, we provide additional implementation details in Appendix A.

**Downstream Tasks.**    The majority of our downstream tasks are performed for image classification. We perform classification on the commonly used datasets ImageNet-1k, ImageNet-100 and CIFAR-100. Additionally, we perform evaluations on images corrupted using 2D Common Corruptions(Hendrycks & Dietterich, 2019), and images perturbed using Adversarial attacks like PGD, APGD (Wong et al., 2020) and AutoAttack.

**Datasets.**    We use the 15 corruptions, with 5 severity levels each, from 2D Common Corruptions(Hendrycks & Dietterich, 2019) (2D-CC), to generate the Common Corruptions version of the respective dataset. We denote this dataset by appending '-C' to the end of the name of the respective dataset, for example, 2D Common Corruptions on ImageNet-1k results into ImageNet-1k-C. We perform our experiments on the following datasets:

**ImageNet-1k**(Russakovsky et al., 2015): This is a subset of the larger ImageNet-22k dataset(Deng et al., 2009), with 1000 object classes, and 1,281,167 training images, 50,000 validation images.

**ImageNet-100**: This is a commonly used(Hoffmann et al., 2021; Tsai et al., 2021; Ge et al., 2021; Lee et al., 2022; Saikia et al., 2021) subset of ImageNet-1k with 100 classes, such that it has 130,000 training images and 5000 validation images, used for faster processing and inference.

**CIFAR-100**(Krizhevsky et al., 2009): This dataset contains 60,000 $32 \times 32$ images, split into 50,000 training images and 10,000 validation images, equally distributed over 100 object classes.

**Adversarial Attacks.** We use PGD(Kurakin et al., 2017), APGD(Wong et al., 2020) and AutoAttack(Croce & Hein, 2020a), with $\ell_\infty$-norm bounded $\epsilon = \frac{4}{255}$ and $\alpha$=0.01 for all our experiments. PGD and APGD are white-box attacks, meaning they require an undisturbed flow of gradients for optimizing their attack. However, AutoAttack, as proposed comprises of APGD-CE (non-targeted APGD attack with Cross Entropy loss), APGD-T (targeted APGD attack), FAB(Croce & Hein, 2020b) and Square(Andriushchenko et al., 2020) Attacks, from these, Square Attack is a black-box attack that does not require a flow of gradients through the ML model. Additionally, since Stochastic Downsampling is essentially a gradient obfuscation method, we also test against Square Attack (Andriushchenko et al., 2020) alone, as it is a black-box attack and does not use the gradient information of a model to optimize the attack.

**Metrics.** For independent and identically distributed (i.i.d.) (non-attacked and non-perturbed) samples, we report the i.i.d. Accuracy (i.i.d. Acc). For evaluations against adversarial attacks, we report the accuracy after the respective attack. For samples from 2D Common Corruptions variant of the respective datasets, we report the mean Corruption Error (mCE), this is the mean error by the method on all the corrupted samples. All numbers are reported in percentages.

**Architectures.** To demonstrate the versatility of Stochastic Downsampling, we have considered multiple architectures and their variants. For experiments with ImageNet-1k, and ImageNet-1k-C, we use ResNet18, ResNet50 and ResNet101 (He et al., 2016), ConvNeXt-T, ConvNeXt-S, and ConvNeXt-B (Liu et al., 2022). We use RobustBench(Croce et al., 2020), to get the adversarially trained weights. For ImageNet-100 and CIFAR-100 (and their 2D-CC counterparts) experiments, we use ConvNeXt-T (tiny) (Liu et al., 2022). Additionally, for comparison, we also use the architectural

changes proposed by (Grabinski et al., 2022a) and (Grabinski et al., 2023), and include them in ConvNeXt-tiny after correspondence with the respective authors.

## A.2 COMPUTE RESOURCES

We used single NVIDIA Tesla V100, and A100 GPUs for each experiment.

## A.3 FINETUNING ON IMAGENET100

We take models pretrained on ImageNet-1k and then finetune them on ImageNet-100 until the training loss converges. This trained model serves as the 'Baseline', and weights from this model are used for finetuning models that are modified with adversarial defense methods. The models with Stochastic Downsampling and other adversarial defenses were trained for 5 epochs for finetuning, with a learning rate of 5e-5 with Cosine Annealing as the learning rate scheduler. We used the SGD optimizer trained using the train split and tested using the test split of the ImageNet100 dataset.

## A.4 FINETUNING ON IMAGENET-1K

The models were trained for 5 epochs for finetuning, with a learning rate of 5e-5 with Cosine Annealing and StepLR as the learning rate scheduler for ConvNeXt and ResNet respectively. We used the SGD optimizer trained using the train split and tested using the test split of the Imagenet-1k dataset.

## A.5 FINETUNING ON CIFAR100

We take models pretrained on ImageNet-1k and then finetune them on CIFAR100 until the training loss converges. This trained model serves as the 'Baseline', and weights from this model are used for finetuning models that are modified with adversarial defense methods. The models were trained for 5 epochs for finetuning, with a learning rate of 4e-4 with Cosine Annealing and MultiStepLR as the learning rate scheduler in case of ConvNeXt and ResNet respectively. We used the SGD optimizer trained using the train split and tested using the test split of the CIFAR100 dataset.

## A.6 TRAINING FROM SCRATCH ON CIFAR100

These models are trained from scratch on CIFAR100. The models were trained for 100 epochs for finetuning, with a learning rate of 0.1 with MultiStepLR as the learning rate scheduler. We used the SGD optimizer trained using the train split and tested using the test split of the CIFAR100 dataset.

# B ADDITIONAL RESULTS

Following we provide additional experimental results and analysis.

## B.1 EXTENDING TO OTHER MODELS AND DATASETS

We extend the analysis from Sec. 5.1 to the CIFAR100 dataset in Tab. 6.

# C CODE FOR STOCHASTIC DOWNSAMPLING

Following is the python code for performing the Stochastic Downsampling operation. It uses pytorch (Paszke et al., 2019).

```python
from typing import Literal, Tuple
import torch
import torch.nn as nn
import torch.nn.functional as F

import einops

class StochasticDownsampler(nn.Module):
```

Table 6: Here we report the performance of various model finetuning strategies against adversarial attacks for CIFAR100. All Methods except 'Baseline' are finetuned for 5 epochs.

| Model | Method | i.i.d. Acc. (%)↑ | PGD Acc. (%)↑ | Autoattack Acc. (%)↑ |
|---|---|---|---|---|
| ConvNeXt-T | Baseline | 81.43 | 2.17 | 0.1 |
| | SD + Addnoise | 68.34 | 30.79 | 19.74 |
| | SD | 72.86 | 19.08 | 6.78 |
| ConvNeXt-S | Baseline | 82.69 | 1.95 | 0.14 |
| | SD + Addnoise | 69.96 | 32.58 | 20.04 |
| | SD | 74.32 | 20.56 | 7.04 |
| ConvNeXt-B | Baseline | 84.45 | 3.45 | 0.18 |
| | SD + Addnoise | 72.46 | 32.79 | 19.94 |
| | SD | 77.05 | 22.58 | 6.38 |
| ResNet18 | Baseline | 76.15 | 1.14 | 0.16 |
| | SD + Addnoise | 73.45 | 1.45 | 0.34 |
| | SD | 75.77 | 1.64 | 0.28 |
| ResNet50 | Baseline | 78.83 | 1.91 | 0.10 |
| | SD + Addnoise | 74.67 | 4.43 | 0.38 |
| | SD | 77.09 | 3.38 | 0.36 |
| ResNet101 | Baseline | 79.83 | 2.06 | 0.08 |
| | SD + Addnoise | 75.25 | 4.74 | 0.32 |
| | SD | 77.79 | 3.44 | 0.34 |

```python
 9      """Stochastically downsamples a feature map to a target resolution. Conceptually approximates a pixel ↩
            integral by monte carlo sampling"""
10      def __init__(self,
11              resolution: Tuple[int, int],
12              spp: int = 16,
13              reduction: Literal["mean", "sum", "min", "max", "prod"] = "mean",
14              jitter_type: Literal["uniform", "normal"] = "uniform",
15              normal_std: float = 1,
16              ):
17          super().__init__()
18          if (not isinstance(resolution, tuple)
19              and not isinstance(resolution, list)):
20              resolution = (resolution, resolution)
21          if len(resolution) != 2:
22              raise ValueError(f"Resolution must be a tuple of length 2, got {resolution}")
23
24          self.resolution = resolution
25          self.spp = spp
26          self.reduction = reduction
27          self.jitter_type = jitter_type
28          self.normal_std = normal_std
29          if self.jitter_type == "uniform":
30              self.jitter_fn = torch.rand
31          elif self.jitter_type == "normal":
32              self.jitter_fn = lambda *args, **kwargs : self.normal_std*torch.randn(*args, **kwargs) + 0.5
33          else:
34              raise NotImplementedError(f"Jitter type {jitter_type} not supported")
35
36      def forward(self, x: torch.Tensor, jitter_array=None):
37          """
38          Downsamples x to the target resolution
39          :param x: high-res input feature map, shape (batch_size, C, H, W)
40          :return: downsampled image, shape (batch_size, C, resolution[0], resolution[1])
41          """
42          b, c, h, w = x.shape
43          resolution, spp = self.resolution, self.spp
44
45          step_x = (1 + 1) / resolution[1]
46          step_y = (1 + 1) / resolution[0]
47          pixel_pos_x = torch.arange(-1, 1, step_x, device=x.device)
48          pixel_pos_y = torch.arange(-1, 1, step_y, device=x.device)
49          pixel_pos = torch.stack(torch.meshgrid(pixel_pos_x, pixel_pos_y, indexing='xy'), dim=2)
50
51          # add subpixel jitter
52          if jitter_array is not None:
53              jitter = jitter_array
54          else:
55              jitter = self.jitter_fn((spp, resolution[0], resolution[1], 2), device=x.device)
56
57          jitter[..., 0] *= step_x
```

```
58        jitter[..., 1] *= step_y
59        pixel_pos = pixel_pos.unsqueeze(0) + jitter # (spp, resolution[0], resolution[1], 2)
60
61        pixel_pos = einops.repeat(pixel_pos, 'spp h w c -> (b spp) h w c', b=b)
62        x_tiled = einops.repeat(x, 'b c h w -> (b spp) c h w', spp=spp)
63
64        samples = F.grid_sample(x_tiled, pixel_pos, mode='bilinear', padding_mode='border', align_corners=↩
              False)
65        return einops.reduce(samples, '(b spp) c h w -> b c h w', self.reduction, b=b)
66
67    def __repr__(self):
68        return (f"StochasticDownsampler(resolution={self.resolution}, "
69               f"spp={self.spp}, reduction='{self.reduction}')")
```

