# OpenReview forum: "Roll the dice: Monte Carlo Downsampling as a low-cost Adversarial Defence"
_ICLR.cc/2025/Conference — ICLR 2025 Conference Withdrawn Submission_

### Official Review · Reviewer_p1uK · 2024-11-01

**Soundness:** 1
**Presentation:** 3
**Contribution:** 1
**Rating:** 1
**Confidence:** 4

**Summary:**

The paper presents a novel empirical defense mechanism against adversarial noise. The authors introduce a new module, the Stochastic Downsampling (SD) Layer, which modifies the stride operation in strided convolutions. This layer decomposes a strided convolution into a non-strided convolution followed by random spatial sampling and bilinear interpolation. The proposed defense aligns with approaches that seek to perturb the gradient, potentially disrupting the attacker’s ability to design an effective adversarial noise against the network.

**Strengths:**

-	The paper is well-written and easy to follow.
-	The method seems easy to implement and lightweight.

**Weaknesses:**

Please find below the weaknesses which I have grouped into major and minor issues.

Major issues:

-    **Experiments are insufficient and do not sufficiently support claims.** In Table 1, the improvement of the proposed method over other methods is low. Moreover, this low improvement is demonstrated only for one model. Broader experiments are required to validate the method. Table 2, despite including more networks, does not offer a comparison with other methods. In Table 3, the proposed method is weaker than the baseline AddNoise (std=0.75), which provides far better robust accuracy while lowly affected by a drop in clean accuracy and mCE.

-   **Results that may required clarifications**.  As explained by the authors, their method might change the direction of the gradients when the attacker designs its attack. If so, I do not understand why an attack designed on a model without SD performs poorly on the same model defended by SD (Table 4). The defended model has been finetuned with the SD module, but I expected that the other layers would not change that much that an attack failed to transfer successfully. Moreover, I do not understand why the Addnoise, which outperforms the SD method in robust defense in Table 3, is now weaker than the SD method in Table 4.

-  **Usefulness of the proposed method.**  I am skeptical about the usefulness of the method over adversarial training. Adversarially trained networks offer better robust accuracy than the SD method (+48%) while not being too much impacted by standard accuracy (-14%) drop. Even in a setting where the defender has a low computational budget, [A] shows that a non-robust network can be quickly adversarially finetuned to obtain robust accuracy. These two points reduce the utility of the proposed method. I think the authors may further discuss and compare their SD method with adversarially trained networks. Moreover, a drawback of the method is that SD cannot be applied to robustify transformer networks while adversarial training can.

Minor issues:

- Ambiguity within the text. At the end of section 3, the authors propose an ablation study named Addnoise, but after Addnoise, it was already referred to as a baseline method. Please clarify this point.

- The network architectures are not introduced in section 4.2 Experimental setup and references for networks are missing.


[A] Jeddi, A., Shafiee, M. J., & Wong, A. (2020). A simple fine-tuning is all you need: Towards robust deep learning via adversarial fine-tuning. arXiv preprint arXiv:2012.13628.

**Questions:**

-	Can you explain why, in Table 5, adding more SPP results in a lower robust accuracy? I would expect that adding more SPP smoothens the use of context and thus pays less attention to local perturbation.

---

### Official Review · Reviewer_JUNr · 2024-11-02

**Soundness:** 2
**Presentation:** 2
**Contribution:** 2
**Rating:** 3
**Confidence:** 4

**Summary:**

The paper proposes Stochastic Downsampling, a Monte Carlo-based downsampling approach designed to add stochasticity to the model's feature maps to help resist adversarial attacks. By combining Monte Carlo integration with bilinear interpolation, this method aims to disrupt adversarial gradients while preserving model performance on clean data. The paper provides experimental results comparing this method to other defense mechanisms across multiple neural architectures and datasets.

**Strengths:**

1. The idea of using Monte Carlo integration for downsampling adds to the diversity of adversarial defense techniques by incorporating randomness directly into feature map transformations.
2. Stochastic Downsampling is computationally efficient compared to adversarial training methods, requiring minimal fine-tuning for integration into existing architectures.

**Weaknesses:**

1. The comparisons in this paper are limited to baseline methods that are not state-of-the-art, which reduces the validity of the findings. To strengthen the evaluation, it is recommended to select and compare against several top-performing methods from the RobustBench leaderboard. Including these advanced and well-established baselines would provide a more meaningful and rigorous benchmark for assessing the effectiveness of the proposed approach.

2. The performance of the proposed method is lower compared to state-of-the-art approaches. For example, on the ImageNet-1k dataset with the ConvNeXt-B architecture, the AutoAttack accuracy is only 3.32%, while robust models on RobustBench can achieve up to 55.82% robust accuracy [1]. This significant discrepancy highlights the method’s limitations in providing strong adversarial robustness.

[1] A Comprehensive Study on Robustness of Image Classification Models: Benchmarking and Rethinking.

3. Given the similarities between the proposed method and gradient confusion techniques, evaluating it with Backward Pass Differentiable Approximation (BPDA) [2] is essential to rule out the possibility that the method relies solely on gradient obfuscation. The absence of BPDA testing leaves unresolved questions about the effectiveness of the proposed method.

[2] Obfuscated gradients give a false sense of security: Circumventing defenses to adversarial examples

4. The presentation of experimental results in the paper lacks completeness, as it does not comprehensively display the performance of the proposed method across all the mentioned datasets.

**Questions:**

Please refer to the weaknesses.

---

### Official Review · Reviewer_gGSq · 2024-11-04

**Soundness:** 2
**Presentation:** 3
**Contribution:** 2
**Rating:** 3
**Confidence:** 4

**Summary:**

This work proposes replacing standard down-sampling operations (e.g., strided convolution) in Deep Neural Networks (DNNs) with stochastic down-sampling as an adversarial defense mechanism. This method requires only a few epochs of fine-tuning to achieve a balance between adversarial robustness and standard accuracy. In the stochastic down-sampling process, some values are sampled using bilinear interpolation, and Monte Carlo integration (stochastic averaging) is performed on these sampled values to calculate the final feature map.

**Strengths:**

This adversarial defense method has noteworthy properties: it is non-parametric, achieves an excellent trade-off between standard accuracy on clean data and adversarial robustness, introduces no extra parameters, and accomplishes this with as few as five epochs. These characteristics make it well-suited for specific use cases.

The paper is generally well-written and easy to follow.

**Weaknesses:**

I find the methodology of the defense mechanism intuitive and well explained. However, I have the following concerns regarding the lack of experimental evaluation to prove its efficacy:

1. While the paper state that stochastic downsampling is a gradient obfuscation-based defense, but it asserts that the technique goes beyond mere gradient obfuscation. To support this claim, the authors evaluated the defense against the Square Attack, which does not require gradient information. However, this evaluation alone does not fully substantiate their claim, as the failure of the Square Attack could be attributed to other factors (e.g., adversarial examples generated through the Square Attack might be highly sensitive to noise and randomness). To demonstrate this point, the authors should include evaluations using EOT[1] that estimates smoother gradients, employing the hyper parameter settings suggested in [3].

2. Although the authors highlighted the limitations of adversarial training (AT) in terms of additional computational cost and clean accuracy degradation, it should not be completely omitted from the evaluation. The authors should have compared the clean accuracy and robustness of adversarial training strategy (e.g, PGD AT, TRADES[2])  with the proposed stochastic downsampling and shown how the computational complexity, adversarial robustness and clean accuracy trade-offs. Furthermore, it is important to consider whether the proposed defense might degrade the robustness of adversarially trained models, as some gradient obfuscation-based methods have been found to do so[3]. Without understanding these trade-offs, the current evaluation of the paper lacks clarity regarding its effectiveness.

3. From table 1, it is evident that the AddNoise benchmark is a strong baseline in terms of clean and adversarial performance compared to the proposed method. Adversarial examples generated by various methods may exhibit differing levels of robustness to noise. To assess how adversarial samples with greater noise robustness perform against the proposed defense method, it is essential to include adversarial examples crafted by Feature Adversaries [4] augmented with EOT[1] in the evaluation.

References:

[1] Athalye, Anish, et al. "Synthesizing robust adversarial examples." International conference on machine learning. PMLR, 2018.

[2] Zhang, Hongyang, et al. "Theoretically principled trade-off between robustness and accuracy." International conference on machine learning. PMLR, 2019.

[3] Tramer, Florian, et al. "On adaptive attacks to adversarial example defenses." Advances in neural information processing systems 33 (2020): 1633-1645.

[4] Sara Sabour, Yanshuai Cao, Fartash Faghri, and David J Fleet. Adversarial manipulation of deep representations. International Conference on Learning Representations,
2016.

**Questions:**

1. The adversarial robustness of the ASAP and Blurpooling benchmarks appears to be quite limited according to Table 1. This is surprising given that the original papers demonstrated some level of robustness. Could there be an issue with the implementation?

2. As outlined in the weaknesses section, additional experimental evaluations are necessary to fully understand the effectiveness of the proposed adversarial defense. I would be happy to raise my score if the concerns are addressed.

---

### Official Review · Reviewer_4pD5 · 2024-11-06

**Soundness:** 2
**Presentation:** 2
**Contribution:** 1
**Rating:** 3
**Confidence:** 5

**Summary:**

This paper proposes Stochastic Downsampling, a Monte Carlo-based downsampling method that enhances neural network robustness against adversarial attacks with minimal impact on clean sample performance. By introducing stochasticity in the downsampling process, it aims to mitigate adversarial perturbations while remaining computationally efficient and easy to integrate into existing architectures.

**Strengths:**

The proposed Stochastic Downsampling technique is a low-cost method that provides robustness against adversarial attacks without requiring additional learnable parameters, making it suitable for deployment in resource-constrained environments​.

**Weaknesses:**

- Missing Related Work on Stochastic Defense: A crucial section on stochastic defense methods is missing in the related work, which leaves out important context for understanding this paper’s contributions relative to existing stochastic defense techniques [1, 2, 3].

- Limited Evaluation of Defense Against Adaptive Attacks: The paper does not thoroughly evaluate the robustness of Stochastic Downsampling against adaptive attacks that might be tailored to circumvent this defense, potentially limiting the method's applicability in adversarial settings.

- The method relies on careful tuning of the number of samples per pixel (SPP) in the Monte Carlo downsampling operation, which could add complexity and potentially limit the method's generalizability across different architectures.

- The evaluations are mainly conducted on smaller datasets like CIFAR-100 and a subset of ImageNet (ImageNet-100), raising questions about the method’s scalability and effectiveness on larger, more complex datasets such as full ImageNet-1k.

- Lack of Comparison with Advanced Defense Methods: The paper lacks comparisons with some of the latest state-of-the-art adversarial defense methods, especially stochastic defenses and noise-based defenses [1,2,3], which would provide better context for evaluating the effectiveness of Stochastic Downsampling relative to other approaches.

- There are several inconsistencies in the reporting of accuracy metrics that could make it challenging for readers to interpret and compare the performance of Stochastic Downsampling effectively. For example, some tables report mean and standard deviation (e.g., i.i.d. accuracy with “mean ± std”), while others omit standard deviation, which can make comparisons across tables unreliable (e.g., Table 1 vs. Table 2 for performance variations on ImageNet100). Also, the trade-off metrics are not clearly defined: some sections report trade-offs between robustness and clean accuracy but lack a consistent metric (e.g., mCE or drop in i.i.d. accuracy) to quantify these trade-offs in a way that can be directly compared across experiments and tables, for which I highly recommend the author read the paper [3].

[1] Liu, Xuanqing, Minhao Cheng, Huan Zhang, and Cho-Jui Hsieh. "Towards robust neural networks via random self-ensemble." In Proceedings of the European Conference on Computer Vision (ECCV), pp. 369-385. 2018.
[2] Wang, X., Wang, S., Chen, P. Y., Lin, X., & Chin, P. (2020, May). Advms: A multi-source multi-cost defense against adversarial attacks. In ICASSP 2020-2020 IEEE International Conference on Acoustics, Speech and Signal Processing (ICASSP) (pp. 2902-2906). IEEE.
[3] Wang, X., Wang, S., Chen, P. Y., Wang, Y., Kulis, B., Lin, X., & Chin, P. (2019). Protecting neural networks with hierarchical random switching: Towards better robustness-accuracy trade-off for stochastic defenses. IJCAI 2019.

**Questions:**

1. How does Stochastic Downsampling compare to other stochastic defense methods in terms of robustness and efficiency [1,2,3]? And how about defense against adaptive attacks? Are there specific reasons why these comparisons were omitted, and could the authors consider including them in future versions?

2. How sensitive is Stochastic Downsampling’s performance to the number of samples per pixel (SPP) parameter? Could the authors provide an analysis of the trade-offs involved in tuning this parameter, especially regarding robustness versus accuracy?

3. The paper primarily focuses on specific architectures (e.g., ConvNeXt-tiny, ResNet18). Have the authors tested Stochastic Downsampling on other architectures, and would it generalize well to models that do not use downsampling layers?

[1] Liu, Xuanqing, Minhao Cheng, Huan Zhang, and Cho-Jui Hsieh. "Towards robust neural networks via random self-ensemble." In Proceedings of the European Conference on Computer Vision (ECCV), pp. 369-385. 2018.
[2] Wang, X., Wang, S., Chen, P. Y., Lin, X., & Chin, P. (2020, May). Advms: A multi-source multi-cost defense against adversarial attacks. In ICASSP 2020-2020 IEEE International Conference on Acoustics, Speech and Signal Processing (ICASSP) (pp. 2902-2906). IEEE.
[3] Wang, X., Wang, S., Chen, P. Y., Wang, Y., Kulis, B., Lin, X., & Chin, P. (2019). Protecting neural networks with hierarchical random switching: Towards better robustness-accuracy trade-off for stochastic defenses. IJCAI 2019.

---

### Author Response · Authors · 2024-11-27
**Thank you for the reviews**

Dear Reviewers,

We sincerely thank all of you for the valuable feedback. We appreciate the effort invested, and we believe incorporating the feedback would significantly improve the quality of our submission.

Unfortunately, revising the submission with additional requested experiments would not be possible in the short discussion phase. Thus, we have chosen to withdraw the submission from consideration at this time. However, we will improve upon it and resubmit it to the next possible venue.

We hope to get reviewers as knowledgeable as you to weigh the paper's evaluations with the revisions.

Best Regards

Authors of Paper #1058

---

### Note · Authors · 2024-11-28

I have read and agree with the venue's withdrawal policy on behalf of myself and my co-authors.